# Non-Stationary Spectral Kernels

**Sami Remes**
sami.remes@aalto.fi

**Markus Heinonen**
markus.o.heinonen@aalto.fi

**Samuel Kaski**
samuel.kaski@aalto.fi

Helsinki Institute for Information Technology HIIT
Department of Computer Science, Aalto University

## Abstract

We propose non-stationary spectral kernels for Gaussian process regression by modelling the spectral density of a non-stationary kernel function as a mixture of input-dependent Gaussian process frequency density surfaces. We solve the generalised Fourier transform with such a model, and present a family of non-stationary and non-monotonic kernels that can learn input-dependent and potentially long-range, non-monotonic covariances between inputs. We derive efficient inference using model whitening and marginalized posterior, and show with case studies that these kernels are necessary when modelling even rather simple time series, image or geospatial data with non-stationary characteristics.

## 1 Introduction

Gaussian processes are a flexible method for non-linear regression [18]. They define a distribution over functions, and their performance depends heavily on the covariance function that constrains the function values. Gaussian processes interpolate function values by considering the value of functions at other similar points, as defined by the kernel function. Standard kernels, such as the Gaussian kernel, lead to smooth neighborhood-dominated interpolation that is oblivious of any periodic or long-range connections within the input space, and can not adapt the similarity metric to different parts of the input space.

Two key properties of covariance functions are *stationarity* and *monotony*. A stationary kernel $K(x, x') = K(x + a, x' + a)$ is a function only of the distance $x - x'$ and not directly the value of $x$. Hence it encodes an identical similarity notion across the input space, while a monotonic kernel decreases over distance. Kernels that are both stationary and monotonic, such as the Gaussian and Matérn kernels, can encode neither input-dependent function dynamics nor long-range correlations within the input space. Non-monotonic and non-stationary functions are commonly encountered in realistic signal processing [19], time series analysis [9], bioinformatics [5, 20], and in geostatistics applications [7, 8].

Recently, several authors have explored kernels that are either non-monotonic or non-stationary. A non-monotonic kernel can reveal informative manifolds over the input space by coupling distant points due to periodic or other effects. Non-monotonic kernels have been derived from the Fourier decomposition of kernels [13, 24, 30], which renders them inherently stationary. Non-stationary kernels, on the other hand, are based on generalising monotonic base kernels, such as the Matérn family of kernels [6, 15], by partitioning the input space [4], or by input transformations [25].

We propose an expressive and efficient kernel family that is – in contrast to earlier methods – both non-stationary and non-monotonic, and hence can infer long-range or periodic relations in an input-dependent manner. We derive the kernel from first principles by solving the more expressive *generalised* Fourier decomposition of non-stationary functions, than the more limited standard Fourier decomposition exploited by earlier works. We propose and solve the generalised spectral density as a mixture of Gaussian process density surfaces that model flexible input-dependent frequency patterns.

The kernel reduces to a stationary kernel with appropriate parameterisation. We show the expressivity of the kernel with experiments on time series data, image-based pattern recognition and extrapolation, and on climate data modelling.

## 2 Related Work

Bochner's theorem for stationary signals, whose covariance can be written as $k(\tau) = k(x - x') = k(x, x')$, implies a Fourier dual [30]

$$k(\tau) = \int S(s)e^{2\pi i s\tau} ds$$

$$S(s) = \int k(\tau)e^{-2\pi i s\tau} d\tau.$$

The dual is a special case of the more general Fourier transform (1), and has been exploited to design rich, yet stationary kernel representations [24, 32] and used for large-scale inference [17]. Lazaro-Gredilla et al. [13] proposed to directly learn the spectral density as a mixture of Dirac delta functions leading to a sparse spectrum (SS) kernel $k_{\text{SS}}(\tau) = \frac{1}{Q} \sum_{i=1}^{Q} \cos(2\pi s_i^T \tau)$.

Wilson et al. [30] derived a stationary spectral mixture (SM) kernel by modelling the univariate spectral density using a mixture of normals $S_{\text{SM}}(s) = \sum_i w_i [\mathcal{N}(s|\mu_i, \sigma_i^2) + \mathcal{N}(s| - \mu_i, \sigma_i^2)]/2$, corresponding to the kernel function $k_{\text{SM}}(\tau) = \sum_i w_i \exp(-2\pi^2 \sigma_i^2 \tau) \cos(2\pi \mu_i \tau)$, which we generalize to the non-stationary case. The SM kernel was also extended for multidimensional inputs using Kronecker structure for scalability [27]. Kernels derived from the spectral representation are particularly well suited to encoding long-range, non-monotonic or periodic kernels; however, they have so far been unable to handle non-stationarity, although [29] presented a partly non-stationary SM kernel that has input-dependent mixture weights. Kom Samo and Roberts also derived a kernel similar to our bivariate spectral mixture kernel in a recent technical report [11].

Non-stationary kernels, on the other hand, have been constructed by non-stationary extensions of Matérn and Gaussian kernels with input-dependent length-scales [3, 6, 15, 16], input space warpings [22, 25], and with local stationarity with products of stationary and non-stationary kernels [2, 23]. The simplest non-stationary kernel is arguably the dot product kernel [18], which has been used as a way to assign input-dependent signal variances [26]. Non-stationary kernels are a good match for functions with transitions in their dynamics, yet are unsuitable for modelling non-monotonic properties.

Our work can also be seen as a generalisation of wavelets, or time-dependent frequency components, into general and smooth input-dependent components. In signal processing, Hilbert-Huang transforms and Hilbert spectral analysis explore input-dependent frequencies, but with deterministic transform functions on the inputs [8, 9].

## 3 Non-stationary spectral mixture kernels

This section introduces the main contributions. We employ the generalised spectral decomposition of non-stationary functions and derive a practical and efficient family of kernels based on non-stationary spectral components. Our approach relies on associating input-dependent frequencies for data inputs, and solving a kernel through the generalised spectral transform.

The most general family of kernels is the non-stationary kernels, which include stationary kernels as special cases [2]. A non-stationary kernel $k(x, x') \in \mathbb{R}$ for scalar inputs $x, x' \in \mathbb{R}$ can be characterized by its spectral density $S(s, s')$ over frequencies $s, s' \in \mathbb{R}$, and the two are related via a generalised Fourier inverse transform[1]

$$k(x, x') = \int_{\mathbb{R}} \int_{\mathbb{R}} e^{2\pi i (xs - x's')} \mu_S(ds, ds') , \qquad (1)$$

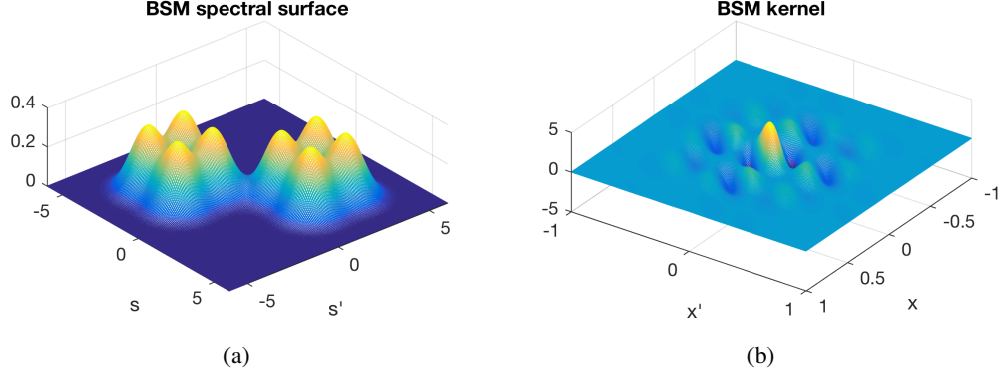

Figure 1: **(a)**: Spectral density surface of a single component bivariate spectral mixture kernel with $8$ permuted peaks. **(b)**: The corresponding kernel on inputs $x \in [-1, 1]$.

where $\mu_S$ is a Lebesgue-Stieltjes measure associated to some positive semi-definite (PSD) spectral density function $S(s, s')$ with bounded variations [2, 14, 31], which we denote as the *spectral surface* since it considers the amplitude of frequency pairs (See Figure 1a).

The generalised Fourier transform (1) specifies that a spectral surface $S(s, s')$ generates a PSD kernel $K(x, x')$ that is non-stationary unless the spectral measure mass is concentrated only on the diagonal $s = s'$. We design a practical, efficient and flexible parameterisation of spectral surfaces that, in turn, specifies novel non-stationary kernels with input-dependent characteristics and potentially long-range non-monotonic correlation structures.

## 3.1 Bivariate Spectral Mixture kernel

Next, we introduce spectral kernels that remove the restriction of stationarity of earlier works. We start by modeling the spectral density as a mixture of $Q$ bivariate Gaussian components

$$S_i(s, s') = \sum_{\boldsymbol{\mu}_i \in \pm\{\mu_i, \mu_i'\}^2} \mathcal{N}\left(\begin{pmatrix} s \\ s' \end{pmatrix} | \boldsymbol{\mu}_i, \Sigma_i\right), \qquad \Sigma_i = \begin{bmatrix} \sigma_i^2 & \rho_i \sigma_i \sigma_i' \\ \rho_i \sigma_i \sigma_i' & \sigma_i'^2 \end{bmatrix}, \qquad (2)$$

with parameterisation using the correlation $\rho_i$, means $\mu_i, \mu_i'$ and variances $\sigma_i^2, \sigma_i'^2$. To produce a PSD spectral density $S_i$ as required by equation (1) we need to include symmetries $S_i(s, s') = S_i(s', s)$ and sufficient diagonal components $S_i(s, s), S_i(s', s')$. To additionally result in a real-valued kernel, symmetry is required with respect to the negative frequencies as well, i.e., $S_i(s, s') = S_i(-s, -s')$. The sum $\sum_{\boldsymbol{\mu}_i \in \pm\{\mu_i, \mu_i'\}^2}$ satisfies all three requirements by iterating over the four permutations of $\{\mu_i, \mu_i'\}^2$ and the opposite signs $(-\mu_i, -\mu_i')$, resulting in eight components (see Figure 1a).

The generalised Fourier inverse transform (1) can be solved in closed form for a weighted spectral surface mixture $S(s, s') = \sum_{i=1}^{Q} w_i^2 S_i(s, s')$ using Gaussian integral identities (see the Supplement):

$$k(x, x') = \sum_{i=1}^{Q} w_i^2 \exp(-2\pi^2 \tilde{\mathbf{x}}^T \Sigma_i \tilde{\mathbf{x}}) \Psi_{\mu_i, \mu_i'}(x)^T \Psi_{\mu_i, \mu_i'}(x') \qquad (3)$$

where

$$\Psi_{\mu_i, \mu_i'}(x) = \begin{pmatrix} \cos 2\pi\mu_i x + \cos 2\pi\mu_i' x \\ \sin 2\pi\mu_i x + \sin 2\pi\mu_i' x \end{pmatrix},$$

and where we define $\tilde{\mathbf{x}} = (x, -x')^T$ and introduce mixture weights $w_i$ for each component. We denote the proposed kernel as the *bivariate spectral mixture* (BSM) kernel (see Figure 1b). The positive definiteness of the kernel is guaranteed by the spectral transform, and is also easily verified since the sinusoidal components form an inner product and the exponential component resembles an unscaled Gaussian density. A similar formulation for non-stationary spectral kernels was presented also in a technical report [11].

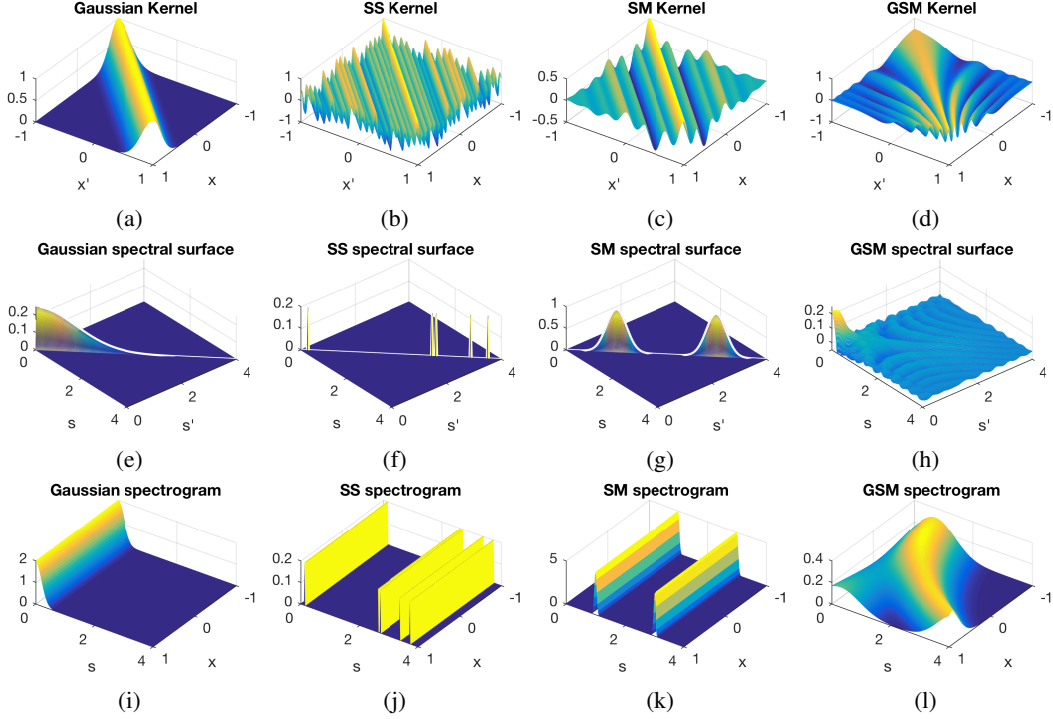

Figure 2: **(a)-(d)**: Examples of kernel matrices on inputs $x \in [-1, 1]$ for a Gaussian kernel (a), sparse spectrum kernel [13] (b), spectral mixture kernel [30] (c), and for the GSM kernel (d). **(e)-(h)**: The corresponding generalised spectral density surfaces of the four kernels. **(i)-(l)**: The corresponding spectrograms, that is, input-dependent frequency amplitudes. The GSM kernel is highlighted with a spectrogram mixture of $Q = 2$ Gaussian process surface functions.

We immediately notice that the BSM kernel vanishes rapidly outside the origin $(x, x') = (0, 0)$. We would require a huge number of components centered at different points $x_i$ to cover a reasonably-sized input space.

## 3.2 Generalised Spectral Mixture (GSM) kernel

We extend the kernel derived in Section 3.1 further by parameterising the frequencies, length-scales and mixture weights as a Gaussian processes[2], that form a smooth spectrogram (See Figure 2(l)):

$$\log w_i(x) \sim \mathcal{GP}(0, k_w(x, x')), \tag{4}$$

$$\log \ell_i(x) \sim \mathcal{GP}(0, k_\ell(x, x')), \tag{5}$$

$$\text{logit}\, \mu_i(x) \sim \mathcal{GP}(0, k_\mu(x, x')). \tag{6}$$

Here the log transform is used to ensure the weights $w(x)$ and lengthscales $\ell(x)$ are non-negative, and the logit transform $\text{logit}\, \mu(x) = \log \frac{\mu}{F_N - \mu}$ limits the learned frequencies between zero and the Nyquist frequency $F_N$, which is defined as half of the sampling rate of the signal.

A GP prior $f(x) \sim \mathcal{GP}(0, k(x, x'))$ defines a distribution over zero-mean functions, and denotes the covariance between function values $\mathbf{cov}[f(x), f(x')] = k(x, x')$ equals their prior kernel. For any collection of inputs, $x_1, \dots, x_N$, the function values follow a multivariate normal distribution $(f(x_1), \dots, f(x_N))^T \sim \mathcal{N}(\mathbf{0}, K)$, where $K_{ij} = k(x_i, x_j)$. The key property of Gaussian processes is that they can encode smooth functions by correlating function values of input points that are similar according to the kernel $k(x, x')$. We use standard Gaussian kernels $k_w$, $k_\ell$ and $k_\mu$.

We accommodate the input-dependent lengthscale by replacing the exponential part of (3) by the Gibbs kernel

$$k_{\text{Gibbs},i}(x, x') = \sqrt{\frac{2\ell_i(x)\ell_i(x')}{\ell_i(x)^2 + \ell_i(x')^2}} \exp\left(-\frac{(x - x')^2}{\ell_i(x)^2 + \ell_i(x')^2}\right) \,,$$

which is a non-stationary generalisation of the Gaussian kernel [3, 6, 15]. We propose a non-stationary *generalised spectral mixture* (GSM) kernel with a simple closed form (see the Supplement):

$$k_{\text{GSM}}(x, x') = \sum_{i=1}^{Q} w_i(x) w_i(x') k_{gibbs,i}(x, x') \cos(2\pi(\mu_i(x)x - \mu_i(x')x')) \,. \tag{7}$$

The kernel is a product of three PSD terms. The GSM kernel encodes the similarity between two data points based on their combined signal variance $w(x)w(x')$, and the frequency surface based on the frequencies $\mu(x), \mu(x')$ and frequency lengthscales $\ell(x), \ell(x')$ associated with both inputs. The GSM kernel encodes the spectrogram surface mixture into a relatively simple kernel. The kernel reduces to the stationary Spectral Mixture (SM) kernel [30] with constant functions $w_i(x) = w_i$, $\mu_i(x) = \mu_i$ and $\ell_i(x) = 1/(2\pi\sigma_i)$ (see the Supplement).

We have presented the proposed kernel (7) for univariate inputs for simplicity. The kernel can be extended to multivariate inputs in a straightforward manner using the generalised Fourier transform with vector-valued inputs [2, 10]. However, in many applications multivariate inputs have a grid-like structure, for instance in geostatistics, image analysis and temporal models. We exploit this assumption and propose a multivariate extension that assumes the inputs to decompose across input dimensions [1, 27]:

$$k_{\text{GSM}}(\mathbf{x}, \mathbf{x}'|\boldsymbol{\theta}) = \prod_{p=1}^{P} k_{\text{GSM}}(x_p, x_p'|\boldsymbol{\theta}_p) \,. \tag{8}$$

Here $\mathbf{x}, \mathbf{x}' \in \mathbb{R}^P$, $\boldsymbol{\theta} = (\boldsymbol{\theta}_1, \ldots, \boldsymbol{\theta}_P)$ collects the dimension-wise kernel parameters $\boldsymbol{\theta}_p = (\mathbf{w}_{ip}, \boldsymbol{\ell}_{ip}, \boldsymbol{\mu}_{ip})_{i=1}^{Q}$ of the $n$-dimensional realisations $\mathbf{w}_{ip}, \boldsymbol{\ell}_{ip}, \boldsymbol{\mu}_{ip} \in \mathbb{R}^n$ per dimension $p$. Then, the kernel matrix can be expressed using Kronecker products as $\mathbf{K}_{\boldsymbol{\theta}} = K_{\boldsymbol{\theta}_1} \otimes \cdots \otimes K_{\boldsymbol{\theta}_P}$, while missing values and data not on a regular grid can be handled with standard techniques [1, 21, 28, 27].

## 4  Inference

We use the Gaussian process regression framework and assume a Gaussian likelihood over $N = n^P$ data points[3] $(\mathbf{x}_j, y_j)_{j=1}^{N}$ with all outputs collected into a vector $\mathbf{y} \in \mathbb{R}^N$,

$$y_j = f(\mathbf{x}_j) + \varepsilon_j, \qquad \varepsilon_j \sim \mathcal{N}(0, \sigma_n^2)$$
$$f(\mathbf{x}) \sim \mathcal{GP}(0, k_{\text{GSM}}(\mathbf{x}, \mathbf{x}'|\boldsymbol{\theta})), \tag{9}$$

with a standard predictive GP posterior $f(\mathbf{x}_\star|\mathbf{y})$ for a new input point $\mathbf{x}_\star$ [18]. The posterior can be efficiently computed using Kronecker identities [21] (see the Supplement).

We aim to infer the noise variance $\sigma_n^2$ and the kernel parameters $\boldsymbol{\theta} = (\mathbf{w}_{ip}, \boldsymbol{\ell}_{ip}, \boldsymbol{\mu}_{ip})_{i=1,p=1}^{Q,P}$ that reveal the input-dependent frequency-based correlation structures in the data, while regularising the learned kernel to penalise overfitting. We perform MAP inference over the log marginalized posterior $\log p(\boldsymbol{\theta}|\mathbf{y}) \propto \log p(\mathbf{y}|\boldsymbol{\theta})p(\boldsymbol{\theta}) = \mathcal{L}(\boldsymbol{\theta})$, where the functions $f(x)$ have been marginalised out,

$$\mathcal{L}(\boldsymbol{\theta}) = \log\left(\mathcal{N}(\mathbf{y}|\mathbf{0}, \mathbf{K}_{\boldsymbol{\theta}} + \sigma_n^2\mathbf{I})\prod_{i,p=1}^{Q,P}\mathcal{N}(\tilde{\mathbf{w}}_{ip}|\mathbf{0}, K_{w_p})\mathcal{N}(\tilde{\boldsymbol{\mu}}_{ip}|\mathbf{0}, K_{\mu_p})\mathcal{N}(\tilde{\boldsymbol{\ell}}_{ip}|\mathbf{0}, K_{\ell_p})\right), \tag{10}$$

where $K_{w_p}, K_{\mu_p}, K_{\ell_p}$ are $n \times n$ prior matrices per dimensions $p$, and $\tilde{\mathbf{w}}, \tilde{\boldsymbol{\mu}}$ and $\tilde{\boldsymbol{\ell}}$ represent the log or logit transformed variables. The marginalized posterior automatically balances between parameters $\boldsymbol{\theta}$ that fit the data and a model that is not overly complex [18]. We can efficiently evaluate both

the marginalized posterior and its gradients in $\mathcal{O}(PN^{\frac{P+1}{P}})$ instead of the usual $\mathcal{O}(N^3)$ complexity [21, 27] (see the Supplement).

Gradient-based optimisation of (10) is likely to converge very slowly due to parameters $\tilde{\mathbf{w}}_{ip}, \tilde{\boldsymbol{\mu}}_{ip}, \tilde{\boldsymbol{\ell}}_{ip}$ being highly self-correlated. We remove the correlations by whitening the variables as $\hat{\boldsymbol{\theta}} = \mathbf{L}^{-1}\tilde{\boldsymbol{\theta}}$ where $\mathbf{L}$ is the Cholesky decomposition of the prior covariances. We maximize $\mathcal{L}$ using gradient ascent with respect to the whitened variables $\hat{\boldsymbol{\theta}}$ by evaluating $\mathcal{L}(\mathbf{L}\hat{\boldsymbol{\theta}})$ and the gradient as [6, 12]

$$\frac{\partial \mathcal{L}}{\partial \hat{\boldsymbol{\theta}}} = \frac{\partial \mathcal{L}}{\partial \boldsymbol{\theta}} \frac{\partial \boldsymbol{\theta}}{\partial \tilde{\boldsymbol{\theta}}} \frac{\partial \tilde{\boldsymbol{\theta}}}{\partial \hat{\boldsymbol{\theta}}} = \mathbf{L}^T \frac{\partial \mathcal{L}}{\partial \tilde{\boldsymbol{\theta}}}. \tag{11}$$

## 5   Experiments

We apply our proposed kernel first on simple simulated time series, then on texture images and lastly on a land surface temperature dataset. With the image data, we compare our method to two stationary mixture kernels, specifically the spectral mixture (SM) [30] and sparse spectrum (SS) kernels [13], and the standard squared exponential (SE) kernel. We employ the GPML Matlab toolbox, which directly implements the SM and SE kernels, and the SS kernel as a meta kernel combining simple cosine kernels. The GPML toolbox also implements Kronecker inference automatically for these kernels.

We implemented the proposed GSM kernel and inference in Matlab[4]. For optimising the log posterior (10) we employ the L-BFGS algorithm. For both our method and the comparisons, we restart the optimisation from 10 different initialisations, each of which is chosen as the best among 100 randomly sampled hyperparameter values as evaluating the log posterior is cheap compared to evaluating gradients or running the full optimisation.

### 5.1   Simulated time series with a decreasing frequency component

First we experiment whether the GSM kernel can find a simulated time-varying frequency pattern. We simulated a dataset where the frequency of the signal changes deterministically as $\mu(x) = 1 + (1-x)^2$ on the interval $x \in [-1, 1]$. We built a single-component GSM kernel $K$ using the specified functions $\mu(x), \ell(x) = \ell = \exp(-1)$ and $w(x) = w = 1$. We sampled a noisy function $\mathbf{y} \sim \mathcal{N}(\mathbf{0}, K + \sigma_n^2 I)$ with a noise variance $\sigma_n^2 = 0.1$. The example in Figure 3 shows the learned GSM kernel, as well as the data and the function posterior $f(x)$. For this 1D case, we also employed the empirical spectrogram for initialising the hyperparameter values. The kernel correctly captures the increasing frequency towards negative values (towards left in Figure 3a).

### 5.2   Image data

We applied our kernel to two texture images. The first image of a sheet of metal represents a mostly stationary periodic pattern. The second, a wood texture, represents an example of a very non-stationary pattern, especially on the horizontal axis. We use majority of the image as training data (the non-masked regions of Figure 3a and 3f) , and use the compared kernels to predict a missing cross-section in the middle, and also to extrapolate outside the borders of the original image.

Figure 4 shows the two texture images, and extrapolation predictions given by the proposed GSM kernel, with a comparison to the spectral mixture (SM), sparse spectrum (SS) and standard squared exponential (SE) kernels. For GSM, SM and SS we used $Q = 5$ mixture components for the metal texture, and $Q = 10$ components for the more complex wood texture.

The GSM kernel gives the most pleasing result visually, and fills in both patterns well with consistent external extrapolation as well. The stationary SM kernel does capture the cross-section, but has trouble extrapolation outside the borders. The SS kernel fails to represent even the training data, it lacks any smoothness in the frequency space. The gaussian kernel extrapolates poorly.

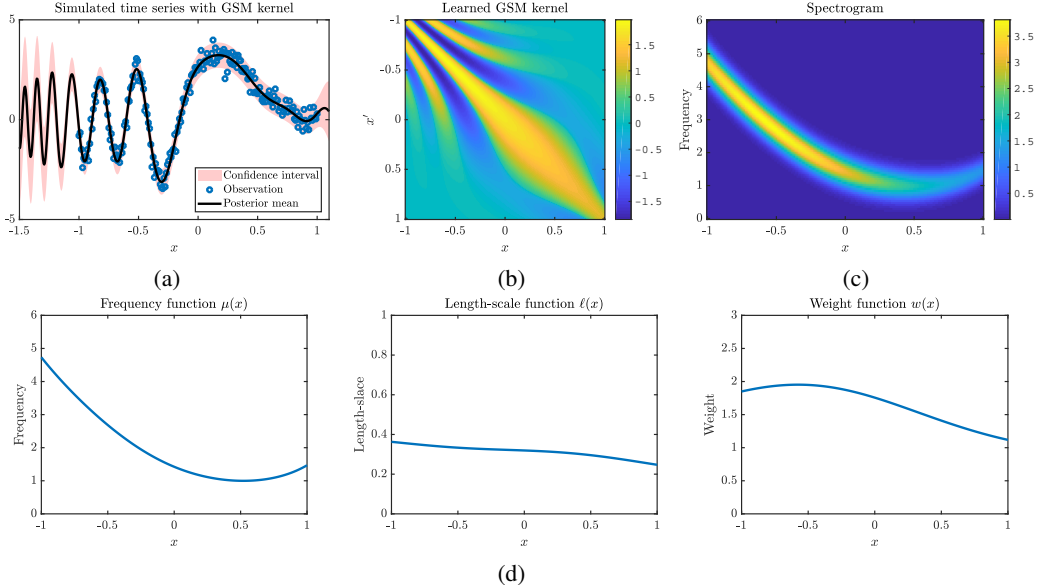

Figure 3: **(a)** A simulated time series with a single decreasing frequency component and a GP fitted using a GSM kernel. **(b)** The learned kernel shows that close to $x = -1$ the signal is highly correlated and anti-correlated with close time points, while these periodic dependencies vanish when moving towards $x = 1$. For visualisation, the values are scaled as $K = \text{sgn}(K)\sqrt{|K|}$. **(c)** The spectrogram shows the decreasing frequency. **(d)** The learned latent frequency function $\mu(x)$ correctly finds the decreasing trend. The length-scale $\ell(x)$ is almost a constant, and weights $w(x)$ slightly decrease in time.

### 5.3 Spatio-Temporal Analysis of Land Surface Temperatures

NASA[5] provides a land surface temperature dataset that we used to demonstrate our kernel in analysis of spatio-temporal data. Our primary objective is to demonstrate the capability of the kernel in inferring long-range, non-stationary spatial and temporal covariances.

We took a subset of four years (February 2000 to February 2004) of North American land temperatures for training data. In total we get 407,232 data points, constituting 48 monthly temperature measurements on a $84 \times 101$ map grid. The grid also contains water regions, which we imputed with the mean temperature of each month. We experimented with the data by learning a generalized spectral mixture kernel using $Q = 5$ components.

Figure 5 presents our results. Figure 5b highlights the training data and model fits for a winter and summer month, respectively. Figure 5a shows the non-stationary kernel slices at two locations across both latitude and longitude, as well as indicating that the spatial covariances are remarkably non-symmetric. Figure 5c indicates five months of successive training data followed by three months of test data predictions.

## 6 Discussion

In this paper we have introduced non-stationary spectral mixture kernels, with treatment based on the generalised Fourier transform of non-stationary functions. We first derived the bivariate spectral mixture (BSM) kernel as a mixture of non-stationary spectral components. However, we argue it has only limited practical use due to requiring an impractical amount of components to cover any sufficiently sized input space. The main contribution of the paper is the generalised spectral mixture (GSM) kernel with input-dependent Gaussian process frequency surfaces. The Gaussian process components can cover non-trivial input spaces with just a few interpretable components. The GSM kernel is a flexible, practical and efficient kernel that can learn both local and global correlations

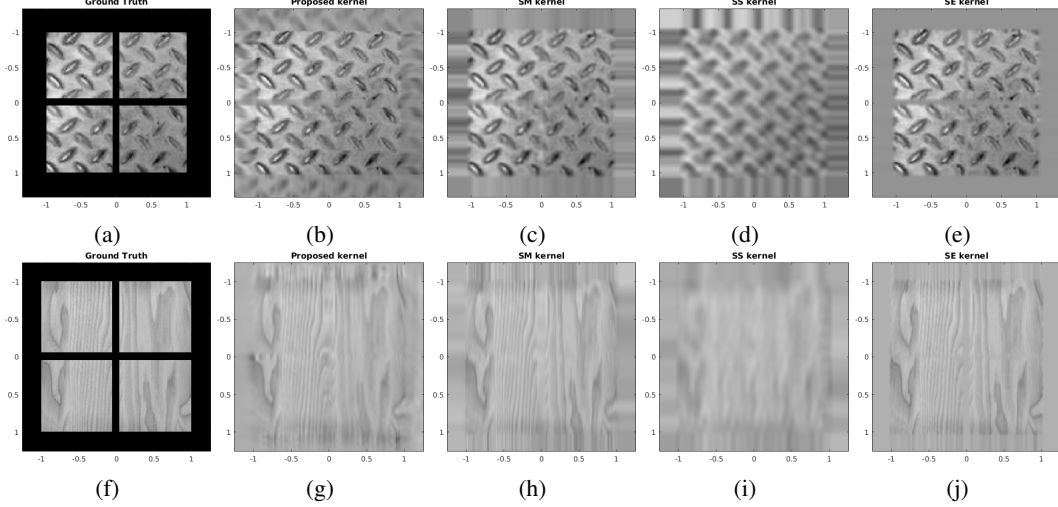

Figure 4: A metal texture data with $Q = 5$ components used for GSM, SM and SS kernels shown in **(a)**-**(e)** and a wood texture in **(f)**-**(j)** (with $Q = 10$ components). The GSM kernel performs the best, making the most believable extrapolation outside image borders in **(b)** and **(g)**. The SM kernel fills in the missing cross pattern in **(c)** but does not extrapolate well. In **(h)** the SM kernel fills in the vertical middle block only with the mean value while GSM in **(g)** is able to fill in a wood-like pattern. SS is not able discover enough structure in either texture **(d)** or **(i)**, while the SE kernel overfits by using a too short length-scale in **(e)** and **(j)**.

across the input domains in an input-dependent manner. We highlighted the capability of the kernel to find interesting patterns in the data by applying it on climate data where it is highly unrealistic to assume the same (stationary) covariance pattern for every spatial location irrespective of spatial structures.

Even though the proposed kernel is motivated by the generalised Fourier transform, the solution to its spectral surface

$$S_{\text{GSM}}(s, s') = \iint k_{\text{GSM}}(x, x') e^{-2\pi i (xs - x's')} dx dx' \tag{12}$$

remains unknown due to having multiple GP functions inside the integral. Figure 2h highlights a numerical integration of the surface equation (12) on an example GP frequency surface. Furthermore, the theoretical work of Kom Samo and Roberts [11] on generalised spectral transforms suggests that the GSM kernel may also be dense in the family of non-stationary kernels, that is, to reproduce arbitrary non-stationary kernels.

### Acknowledgments

This work has been partly supported by the Finnish Funding Agency for Innovation (project Re:Know) and Academy of Finland (COIN CoE, and grants 299915, 294238 and 292334). We acknowledge the computational resources provided by the Aalto Science-IT project.

## Footnotes

[1]We focus on scalar inputs and frequencies for simplicity. An extension based on vector-valued inputs and frequencies [2, 10] is straightforward.

[2]See the Supplement for a tutorial on Gaussian processes.

[3]Assuming that we have equal number of points $n$ in all dimensions.

[4]Implementation available at `https://github.com/sremes/nonstationary-spectral-kernels`

[5]`https://neo.sci.gsfc.nasa.gov/view.php?datasetId=MOD11C1_M_LSTDA`

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
