[Supplementary Material · Supplementary_Spectral_Kernels.pdf]

# Supplementary material:
# Non-stationary Spectral Kernels

November 3, 2017

## 1  A tutorial on Gaussian processes

We summarise here Gaussian process regression for completeness. For an interested reader, we refer to the excellent and comprehensive book by Rasmussen and Williams [5].

Gaussian processes (GP) are a Bayesian nonparameteric machine learning framework for regression, classification and unsupervised learning [5]. A Gaussian process is a collection of random variables, any finite combination of which has a Multivariate normal distribution. A GP prior defines a distribution over functions, denoted as

$$f(x) \sim \mathcal{GP}(m(x), k(x, x')), \tag{1}$$

where the mean function $m(x)$ and a positive semi-definite kernel function $K(x, x')$ for inputs $x \in \mathbb{R}$ determine the function expectation and covariance,

$$\mathbf{E}[f(x)] = m(x) \tag{2}$$
$$\mathbf{cov}[f(x), f(x')] = k(x, x'). \tag{3}$$

Furthermore, the GP prior determines that for any finite collection of input points $x_1, \ldots, x_N$, the corresponding function values follow a Multivariate normal distribution

$$p(f(x_1), \ldots, f(x_N)) \sim \mathcal{N}(\mathbf{m}, K), \tag{4}$$

where $\mathbf{m} = (m(x_1), \ldots, m(x_N))^T \in \mathbb{R}^N$, and $K \in \mathbb{R}^{N \times N}$ with $K_{ij} = k(x_i, x_j)$. A Gaussian process models functions where for similar points $x, x'$ their corresponding function values $f(x), f(x')$ are also similar. A common kernel choice is the Gaussian kernel

$$k(x, x') = \sigma_f^2 \exp\left(-\frac{1}{2}\frac{(x - x')^2}{\ell^2}\right), \tag{5}$$

which encodes monotonic neighborhood similarity. The kernel parameters are the signal variance $\sigma_f^2$ and the kernel lengthscale $\ell$.

Assume a dataset $\mathcal{D} = (x_i, y_i)_{i=1}^N$ and an additive Gaussian likelihood

$$y = f(x) + \varepsilon(x), \qquad \varepsilon(x) \sim \mathcal{N}(0, \sigma_n^2) \tag{6}$$

with a data likelihood

$$p(\mathbf{y}|\mathbf{f}) = \mathcal{N}(\mathbf{y}|\mathbf{f}, \sigma_n^2 I), \tag{7}$$

where $\mathbf{y} = (y_1, \ldots, y_N)^T \in \mathbb{R}^N$ collects the observed outputs corresponding to inputs $(x_1, \ldots, x_N)$, and $\mathbf{f} = (f(x_1), \ldots, f(x_N))^T \in \mathbb{R}^N$ collects the function values, and $\sigma_n^2$ is the noise variance. The predictive distribution of $f(x_*)|\mathbf{y}$ for a new point $x_*$ conditioned on the data $\mathbf{y}$ at training inputs $X$ is again a Gaussian

$$f(x_*)|\mathbf{y} \sim \mathcal{N}(\mu_*, \sigma_*^2) \tag{8}$$

$$\mu_* = K(x_*, X)(K + \sigma_n^2)^{-1}(\mathbf{y} - \mathbf{m}) + \mathbf{m} \tag{9}$$

$$\sigma_*^2 = K(x_*, x_*) - K(x_*, X)(K + \sigma_n^2)^{-1}K(X, x_*), \tag{10}$$

where $K(x_*, X) = K(X, x_*)^T$ is a row kernel.

Since the full predictive distribution is in closed form, the inference task is shifted to learning the hyperparameters $\theta = (\sigma_f, \ell, \sigma_n)$. The log marginalized likelihood

$$\log p(\mathbf{y}|\theta) = \log \int p(\mathbf{y}|\mathbf{f})p(\mathbf{f}|\theta)d\mathbf{f} \tag{11}$$

$$= \log \int \mathcal{N}(\mathbf{y}|\mathbf{f}, \sigma_n^2 I)\mathcal{N}(\mathbf{f}|\mathbf{m}, K)d\mathbf{f} \tag{12}$$

$$= \log \mathcal{N}(\mathbf{y}|\mathbf{m}, K + \sigma_n^2 I) \tag{13}$$

$$\propto -\frac{1}{2}(\mathbf{y} - \mathbf{m})^T(K + \sigma_n^2)^{-1}(\mathbf{y} - \mathbf{m}) - \frac{1}{2}\log|K + \sigma_n^2| \tag{14}$$

has a closed form as well. The marginal log likelihood is related to the amount of functions compatible with the prior and matching the data. Hence, the marginal log likelihood automatically promotes priors that induce functions matching the data while penalising model complexity. The marginal log likelihood can be directly maximised using standard gradient ascent techniques to infer optimal hyperparameters $\theta$.

## 2 Deriving the bivariate spectral mixture kernel

A non-stationary kernel $k(x, x') \in \mathbb{R}$ for scalar inputs $x, x' \in \mathbb{R}$ can characterized by its spectral density $S(s, s')$ over frequencies $s, s' \in \mathbb{R}$, and the two are related via a generalised Fourier transform [10, 4]

$$k(x, x') = \int_{\mathbb{R}} \int_{\mathbb{R}} e^{2\pi i(xs - x's')}\mu_S(ds, ds') \tag{15}$$

where $\mu_S$ is a Lebesgue-Stieltjes measure associated to some positive semi-definite (PSD) spectral density function $S(s, s')$ with bounded variations, which we denote as the *spectral surface* since it considers the amplitude of frequency pairs.

We define a spectral density $S(s, s')$ as a mixture of $Q$ bivariate Gaussian components

$$S_i(s, s') = \sum_{\boldsymbol{\mu}_i \in \pm\{\mu_i, \mu_i'\}^2} \mathcal{N}\left(\begin{pmatrix} s \\ s' \end{pmatrix} | \boldsymbol{\mu}_i, \Sigma_i\right) \tag{16}$$

$$\Sigma_i = \begin{bmatrix} \sigma_i^2 & \rho_i \sigma_i \sigma_i' \\ \rho_i \sigma_i \sigma_i' & \sigma_i'^2 \end{bmatrix}$$

with parameterization using the correlation $\rho_i$, means $\mu_i, \mu_i'$ and variances $\sigma_i^2, \sigma_i'^2$. To ensure the PSD property of spectral density $S_i(s, s')$ it must hold that $S_i(s, s') = S_i(s', s)$ and sufficient diagonal components $S_i(s, s)$, $S_i(s', s')$ exist. In addition to retrieve a real-valued kernel we require symmetry with respect to the negative frequencies as well, i.e. $S_i(s, s') = S_i(-s, -s')$. The sum $\sum_{\boldsymbol{\mu}_i \in \pm\{\mu_i, \mu_i'\}^2}$ satisfies all three requirements by iterating over four permutations of $\{\mu_i, \mu_i'\}^2$ and the opposite signs $(-\mu_i, -\mu_i')$, resulting in eight components

$$\pm\{\mu, \mu'\}^2 = \{(\mu, \mu), (\mu, \mu'), (\mu', \mu), (\mu', \mu'), (-\mu, -\mu), (-\mu, -\mu'), (-\mu', -\mu), (-\mu', -\mu')\}.$$

The full $Q$-component spectral density is

$$S(s, s') = \sum_{i=1}^{Q} \sum_{\boldsymbol{\mu}_i \in \pm\{\mu_i, \mu_i'\}^2} \mathcal{N}\left(\begin{pmatrix} s \\ s' \end{pmatrix} | \boldsymbol{\mu}_i, \Sigma_i\right). \tag{17}$$

Next, we compute the generalised Fourier transform in closed form by exploiting Gaussian integral identities

$$k(x, x') = \int_{\mathbb{R}} \int_{\mathbb{R}} S(s, s') e^{2\pi i(xs - x's')} ds ds' \tag{18}$$

$$= \int_{\mathbb{R} \times \mathbb{R}} \sum_{i=1}^{Q} \sum_{\boldsymbol{\mu}_i \in \pm\{\mu_i, \mu_i'\}^2} \mathcal{N}\left(\begin{pmatrix} s \\ s' \end{pmatrix} | \boldsymbol{\mu}_i, \Sigma_i\right) e^{2\pi i \tilde{\mathbf{x}}^T \mathbf{s}} d\mathbf{s} \tag{19}$$

$$= \sum_{i=1}^{Q} \sum_{\boldsymbol{\mu}_i \in \pm\{\mu_i, \mu_i'\}^2} \int_{\mathbb{R} \times \mathbb{R}} \mathcal{N}(\mathbf{s} | \boldsymbol{\mu}_i, \Sigma_i) e^{2\pi i \tilde{\mathbf{x}}^T \mathbf{s}} d\mathbf{s} \tag{20}$$

$$= \sum_{i=1}^{Q} \sum_{\boldsymbol{\mu}_i \in \pm\{\mu_i, \mu_i'\}^2} \frac{1}{(2\pi)^2 |\Sigma_i|} \int \exp\left(-\frac{1}{2}(\mathbf{s} - \boldsymbol{\mu}_i)^T \Sigma_i^{-1}(\mathbf{s} - \boldsymbol{\mu}_i) + \mathbf{b}^T \mathbf{s}\right) d\mathbf{s} \tag{21}$$

$$= \sum_{i=1}^{Q} \sum_{\boldsymbol{\mu}_i \in \pm\{\mu_i, \mu_i'\}^2} \frac{w^2}{(2\pi)^2 |\Sigma_i|} \int \exp\left(-\frac{1}{2}\mathbf{s}^T \Sigma_i^{-1} \mathbf{s} + (\mathbf{b} + \Sigma_i^{-1} \boldsymbol{\mu}_i)^T \mathbf{s} - \frac{1}{2}\boldsymbol{\mu}_i^T \Sigma_i^{-1} \boldsymbol{\mu}_i\right) d\mathbf{s} \tag{22}$$

$$= \sum_{i=1}^{Q} \sum_{\boldsymbol{\mu}_i \in \pm\{\mu_i, \mu_i'\}^2} \exp\left(\frac{1}{2}(\mathbf{b} + \Sigma_i^{-1} \boldsymbol{\mu}_i)^T \Sigma_i (\mathbf{b} + \Sigma_i^{-1} \boldsymbol{\mu}_i)\right) \exp\left(-\frac{1}{2}\boldsymbol{\mu}_i^T \Sigma_i^{-1} \boldsymbol{\mu}_i\right) \tag{23}$$

$$= \sum_{i=1}^{Q} \sum_{\boldsymbol{\mu}_i \in \pm\{\mu_i, \mu_i'\}^2} \exp\left(\frac{1}{2}\mathbf{b}^T \Sigma_i \mathbf{b} + \boldsymbol{\mu}_i^T \mathbf{b}\right) \tag{24}$$

where we defined $\tilde{\mathbf{x}} = (x, -x')^T$ and $\mathbf{s} = (s, s')^T$, and $\mathbf{b} = (2\pi i x, -2\pi i x')^T$.

The $i$'th component of the kernel mixture is then

$$k_i(x, x') = e^{-2\pi^2 \tilde{\mathbf{x}}^T \Sigma \tilde{\mathbf{x}}} [ \quad e^{2\pi i \mu x} e^{-2\pi i \mu' x'} + e^{2\pi i \mu' x} e^{-2\pi i \mu x'} + e^{2\pi i \mu x} e^{-2\pi i \mu x'} + e^{2\pi i \mu' x} e^{-2\pi i \mu' x'} \tag{25}$$

$$+ e^{-2\pi i \mu x} e^{2\pi i \mu' x'} + e^{-2\pi i \mu' x} e^{2\pi i \mu x'} + e^{-2\pi i \mu x} e^{2\pi i \mu x'} + e^{-2\pi i \mu' x} e^{2\pi i \mu' x'}]$$

which can be simplified by noting that

$$e^{2\pi i \mu x} e^{-2\pi i \mu' x'} + e^{-2\pi i \mu x} e^{2\pi i \mu' x'}$$
$$= (\cos(2\pi\mu x) + i \sin(2\pi\mu x))(\cos(2\pi\mu' x') - i \sin(2\pi\mu' x'))$$
$$+ (\cos(2\pi\mu x) - i \sin(2\pi\mu x))(\cos(2\pi\mu' x') + i \sin(2\pi\mu' x'))$$
$$= 2\cos(2\pi\mu x)\cos(2\pi\mu' x') + 2\sin(2\pi\mu x)\sin(2\pi\mu' x')$$

where the complex part cancels out. Now by defining a function

$$\Psi_{\mu, \mu'}(x) = \begin{pmatrix} \cos 2\pi\mu x + \cos 2\pi\mu' x \\ \sin 2\pi\mu x + \sin 2\pi\mu' x \end{pmatrix} \tag{26}$$

we can express the sum of the 8 exponentials in (25) as $\Psi_{\mu, \mu'}(x)^T \Psi_{\mu, \mu'}(x')$. The final kernel thus takes the form

$$k(x, x') = \sum_{i=1}^{Q} w_i^2 e^{-2\pi^2 \tilde{\mathbf{x}}^T \Sigma_i \tilde{\mathbf{x}}} \Psi_{\mu_i, \mu_i'}(x)^T \Psi_{\mu_i, \mu_i'}(x'), \tag{27}$$

where we introduced mixture weights $w_i$ for each component.

Now, we immediately notice that the kernel vanishes rapidly outside the origin $(x, x') = (0, 0)$; we would require a huge number of components centered at different points $x_i$ to cover a reasonably-sized input space. One simple fix would be to change the exponential part to e.g. a Gaussian kernel $\exp(-\frac{1}{2}\sigma^2||x - x'||^2)$ to prevent the component from vanishing but this still would not allow us to account for non-stationary frequencies, which is what we address next.

# 3 Deriving the generalised spectral mixture (GSM) kernel

The generalised spectral mixture kernel defines Gaussian process frequencies, lengthscales and mixture weights:

$$\log w_i(x) \sim \mathcal{GP}(0, k_w(x, x')), \tag{28}$$
$$\log \ell_i(x) \sim \mathcal{GP}(0, k_\ell(x, x')), \tag{29}$$
$$\operatorname{logit} \mu_i(x) \sim \mathcal{GP}(0, k_\mu(x, x')), \tag{30}$$

where we use the log transform to ensure weights $w(x)$ and lengthscales $\ell(x)$ are positive, and we use the logit transformed. The transform $\hat{\mu}$ and the inverse transform $\mu$ is given by

$$\operatorname{logit}(\mu) = \hat{\mu} = \log \frac{\mu}{F_N - \mu} \tag{31}$$

$$\mu = \frac{F_N}{1 + \exp(-\hat{\mu})}. \tag{32}$$

Frequency parameter $\operatorname{logit} \mu(x)$ to limit the learned frequencies between zero and the Nyquist frequency $F_N$, which can be defined as half of the sampling rate of the signal (or for non-equispaced signals as the inverse of the smallest time interval between the samples).

To accommodate lengthscale functions we replace the exponential part of the BSM kernel by the Gibbs kernel

$$k_{gibbs,i}(x, x') = \sqrt{\frac{2\ell_i(x)\ell_i(x')}{\ell_i(x)^2 + \ell_i(x')^2}} \exp\left(-\frac{(x - x')^2}{\ell_i(x)^2 + \ell_i(x')^2}\right).$$

The cosine part (26) is replaced by a function

$$\Psi_i(x) = \begin{pmatrix} \cos(2\pi\mu_i(x)x) \\ \sin(2\pi\mu_i(x)x) \end{pmatrix}.$$

The non-stationary *generalised spectral mixture* (GSM) kernel has a closed form

$$k_{gsm}(x, x') = \sum_{i=1}^{Q} w_i(x)w_i(x')k_{gibbs}(x, x')\Psi_i(x)^T\Psi_i(x') \tag{33}$$

$$= \sum_{i=1}^{Q} w_i(x)w_i(x')k_{gibbs,i}(x, x')\cos(2\pi(\mu_i(x)x - \mu_i(x')x')) \tag{34}$$

due to identity $\cos\alpha\cos\beta + \sin\alpha\sin\beta = \cos(\alpha - \beta)$. The kernel is a product of three kernels, namely a linear kernel, a Gibbs kernel and a novel cosine kernel with a feature mapping $\Psi_i(x)$. The full kernel is PSD due to all of its product kernels being PSD. The cosine kernel is PSD due to a dot product.

## 3.1 Relationship between Spectral Mixture kernel and the Generalised Spectral Mixture kernel

We show that the proposed non-stationary GSM kernel reduces to the stationary SM kernel with appropriate parameterisation. We show this identity for univariate inputs for simplicity, with the same result being straightforward to derive for multivariate kernel variants as well.

The proposed generalised spectral mixture (GSM) kernel for univariate inputs is

$$k_{\text{GSM}}(x,x') = \sum_{i=1}^{Q} w_i(x)w_i(x') \sqrt{\frac{2\ell_i(x)\ell_i(x')}{\ell_i(x)^2 + \ell_i(x')^2}} \exp\left(-\frac{(x-x')^2}{\ell_i(x)^2 + \ell_i(x')^2}\right) \cos\left(2\pi(\mu_i(x)x - \mu_i(x')x')\right) \quad (35)$$

with Gaussian process functions $w_i(x), \mu_i(x), \ell_i(x)$. The Spectral Mixture (SM) kernel by Wilson et al [9] is

$$k_{\text{SM}}(x,x') = \sum_{i=1}^{Q} w_i^2 \exp(-2\pi^2(x-x')^2\sigma_i^2)\cos(2\pi\mu_i(x-x')) \quad (36)$$

$$S_{\text{SM}}(s) = \sum_{i=1}^{Q} w_i^2 \left[\mathcal{N}(s|\mu_i,\sigma_i^2) + \mathcal{N}(s|-\mu_i,\sigma_i^2)\right], \quad (37)$$

where the parameters are the weights $w_i$, mean frequencies $\mu_i$ and variances $\sigma_i^2$. Now if we assign the following constant functions for the GSM kernel to match the parameters of the SM kernel on the right-hand side,

$$w_i(x) = w_i \quad (38)$$

$$\mu_i(x) = \mu_i \quad (39)$$

$$\ell_i(x) = \frac{1}{2\pi\sigma_i}, \quad (40)$$

we retrieve the SM kernel

$$k_{\text{GSM}}(x,x') = \sum_{i=1}^{Q} w_i(x)w_i(x') \sqrt{\frac{2\ell_i(x)\ell_i(x')}{\ell_i(x)^2 + \ell_i(x')^2}} \exp\left(-\frac{(x-x')^2}{\ell_i(x)^2 + \ell_i(x')^2}\right) \cos(2\pi(\mu_i(x)x - \mu_i(x')x')) \quad (41)$$

$$= \sum_{i=1}^{Q} w_i^2 \exp\left(-\frac{(x-x')^2}{2(1/(2\pi\sigma_i))^2}\right) \cos(2\pi\mu(x-x')) \quad (42)$$

$$= \sum_{i=1}^{Q} w_i^2 \exp\left(-\frac{1}{2}(2\pi\sigma_i)^2(x-x')^2\right) \cos(2\pi\mu(x-x')) \quad (43)$$

$$= \sum_{i=1}^{Q} w_i^2 \exp\left(-2\pi^2\sigma_i^2(x-x')^2\right) \cos(2\pi\mu(x-x')) \quad (44)$$

$$= k_{\text{SM}}(x,x'). \quad (45)$$

This indicates that the GSM kernel can reproduce any kernel that is reproducable by the SM kernel, which is known to be a highly flexible kernel [9, 8]. In practise we can simulate stationary kernels by setting the spectral function priors $k_w, k_\mu, k_\ell$ to enforce very smooth, or in practise constant, functions.

# 4 Inference

In many applications multivariate inputs have a grid-like structure, for instance in geostatistics, image analysis and temporal models. We exploit this assumption and propose a multivariate extension that assumes

the inputs to decompose across input dimensions [1, 9]:

$$k_{\mathrm{GSM}}(\mathbf{x}, \mathbf{x}'|\boldsymbol{\theta}) = \prod_{p=1}^{P} k_{\mathrm{GSM}}(x_p, x_p'|\boldsymbol{\theta}_p) \,. \tag{46}$$

Here $\mathbf{x}, \mathbf{x}' \in \mathbb{R}^P$, $\boldsymbol{\theta} = (\boldsymbol{\theta}_1, \ldots, \boldsymbol{\theta}_P)$ collects the dimension-wise kernel parameters $\boldsymbol{\theta}_p = (\mathbf{w}_{ip}, \boldsymbol{\ell}_{ip}, \boldsymbol{\mu}_{ip})_{i=1}^{Q}$ of the $N$-dimensional realisations $\mathbf{w}_{ip}, \boldsymbol{\ell}_{ip}, \boldsymbol{\mu}_{ip} \in \mathbb{R}^N$ per dimension $p$. Then, the kernel matrix can be expressed using Kronecker products as $\mathbf{K}_{\boldsymbol{\theta}} = K_{\boldsymbol{\theta}_1} \otimes \cdots \otimes K_{\boldsymbol{\theta}_P}$, while missing values and data not on a regular grid can be handled with standard techniques [1, 6, 8, 7].

We use the Gaussian process regression framework and assume a Gaussian likelihood over $N^P$ data points $(\mathbf{x}_j, y_j)_{j=1}^{N^P}$ with all outputs collected into a vector $\mathbf{y} \in \mathbb{R}^{N^P}$,

$$y_j = f(\mathbf{x}_j) + \varepsilon_j, \qquad \varepsilon_j \sim \mathcal{N}(0, \sigma_n^2)$$
$$f(\mathbf{x}) \sim \mathcal{GP}(0, k_{\mathrm{GSM}}(\mathbf{x}, \mathbf{x}'|\boldsymbol{\theta})), \tag{47}$$

with a standard predictive GP posterior $f(\mathbf{x}_\star|\mathbf{y})$ for a new input point $\mathbf{x}_\star$ [5]. The posterior can be efficiently computed using Kronecker identities [6].

We aim to infer the noise variance $\sigma_n^2$ and the kernel parameters $\boldsymbol{\theta} = (\mathbf{w}_{ip}, \boldsymbol{\ell}_{ip}, \boldsymbol{\mu}_{ip})_{i=1,p=1}^{Q,P}$ that reveal the input-dependent frequency-based correlation structures in the data, while regularising the learned kernel to penalise overfitting. We perform MAP inference over the log marginalized posterior $\log p(\boldsymbol{\theta}|\mathbf{y}) \propto \log p(\mathbf{y}|\boldsymbol{\theta}) p(\boldsymbol{\theta}) = \mathcal{L}(\boldsymbol{\theta})$, where the functions $f(x)$ have been marginalised out,

$$\mathcal{L}(\boldsymbol{\theta}) = \log \left( \mathcal{N}(\mathbf{y}|\mathbf{0}, \mathbf{K}_{\boldsymbol{\theta}} + \sigma_n^2 \mathbf{I}) \prod_{i,p=1}^{Q,P} \mathcal{N}(\mathbf{w}_{ip}|\mathbf{0}, K_{w_p}) \mathcal{N}(\boldsymbol{\mu}_{ip}|\mathbf{0}, K_{\mu_p}) \mathcal{N}(\boldsymbol{\ell}_{ip}|\mathbf{0}, K_{\ell_p}) \right) \tag{48}$$

$$\propto -\mathbf{y}^T (\mathbf{K}_{\boldsymbol{\theta}} + \sigma^2 I)^{-1} \mathbf{y} - \log |\mathbf{K}_{\boldsymbol{\theta}} + \sigma_n^2 I|$$

$$- \sum_{p=1}^{P} \sum_{i=1}^{Q} \left( \mathbf{w}_{ip}^T K_{w_p}^{-1} \mathbf{w}_{ip} - \boldsymbol{\ell}_{ip}^T K_{\ell_p}^{-1} \boldsymbol{\ell}_{ip} - \boldsymbol{\mu}_{ip}^T K_{\mu_p}^{-1} \boldsymbol{\mu}_{ip} \right) - Q \sum_{p=1}^{P} \left( \log |K_{w_p}| - \log |K_{\ell_p}| - \log |K_{\mu_p}| \right)$$

where $K_{w_p}, K_{\mu_p}, K_{\ell_p}$ are $N \times N$ prior matrices per dimensions $p$. The marginalized posterior automatically balances between parameters $\boldsymbol{\theta}$ that fit the data and a model that is not overly complex [5]. We can efficiently evaluate both the marginalized posterior and its gradients in $\mathcal{O}(PN^{\frac{P+1}{P}})$ instead of the usual $\mathcal{O}(N^{P^3})$ complexity [6] (See Supplements).

Gradient-based optimisation of (48) is likely to converge very slowly due to parameters $\mathbf{w}_{ip}, \boldsymbol{\mu}_{ip}, \boldsymbol{\ell}_{ip}$ being highly self-correlated. We remove the correlations by whitening the variables as $\tilde{\boldsymbol{\theta}} = \mathbf{L}^{-1} \boldsymbol{\theta}$ where $\mathbf{L}$ is the Cholesky decomposition of the prior covariances. We maximize $\mathcal{L}(\boldsymbol{\theta})$ using gradient ascent with respect to the whitened variables $\tilde{\boldsymbol{\theta}}$ by evaluating $\mathcal{L}(\mathbf{L}\tilde{\boldsymbol{\theta}})$ and the gradient as [3, 2]

$$\frac{\partial \mathcal{L}(\boldsymbol{\theta})}{\partial \tilde{\boldsymbol{\theta}}} = \frac{\partial \mathcal{L}(\boldsymbol{\theta})}{\partial \boldsymbol{\theta}} \frac{\partial \boldsymbol{\theta}}{\partial \tilde{\boldsymbol{\theta}}} = \mathbf{L}^T \frac{\partial \mathcal{L}(\boldsymbol{\theta})}{\partial \boldsymbol{\theta}}. \tag{49}$$

## 4.1 Kronecker inference

The marginal likelihood (48) can be evaluated using the eigen decomposition $\mathbf{K} = \boldsymbol{Q} \boldsymbol{V} \boldsymbol{Q}^T$. Using known results for Kronecker products we can compute the eigen decomposition as $\boldsymbol{Q} = \bigotimes_p Q_p$, $\boldsymbol{V} = \bigotimes_p V_p$ and $\boldsymbol{Q}^T = \bigotimes_p Q_p^T$ using the decompositions of the smaller kernels $K_p = Q_p V_p Q_p^T$. Thus we can decompose the computation of the first term in (48) as

$$(\mathbf{K} + \sigma_n^2 \mathbf{I})^{-1} \mathbf{y} = \boldsymbol{Q} (\boldsymbol{V} + \sigma_n^2 \mathbf{I})^{-1} \boldsymbol{Q}^T \mathbf{y} = \left( \bigotimes_p Q_p \right) \left( (\boldsymbol{V} + \sigma_n^2 \mathbf{I})^{-1} \left( \left( \bigotimes_p Q_p^T \right) \mathbf{y} \right) \right), \tag{50}$$

where the inversion is taken only of the diagonal matrix of eigenvalues and matrix-vector products with a Kronecker matrix can be computed efficiently. The second term of (48) can be computed using the eigenvalues $\boldsymbol{\lambda} = \text{diag}(\boldsymbol{V}) = \bigotimes_p \text{diag}(V_p)$ as $\log |\mathbf{K} + \sigma_n^2 \mathbf{I}| = \sum_i \log(\lambda_i + \sigma_n^2)$.

The gradient of the marginal likelihood is given by

$$\frac{\partial \mathcal{L}}{\partial \theta_p} = \frac{1}{2} \left( \boldsymbol{\alpha}^T \frac{\partial \mathbf{K}}{\partial \theta_p} \boldsymbol{\alpha} - \text{tr} \left( (\mathbf{K} + \sigma_n^2 \mathbf{I})^{-1} \frac{\partial \mathbf{K}}{\partial \theta_p} \right) \right), \tag{51}$$

where $\boldsymbol{\alpha} = (\mathbf{K} + \sigma_n^2 \mathbf{I})^{-1} \mathbf{y}$ is computed as in (50). The gradient of the Kronecker product kernel can be computed as

$$\frac{\partial \mathbf{K}}{\partial \theta_p} = K_1 \otimes \ldots \otimes \frac{\partial K_p}{\partial \theta_p} \otimes \ldots \otimes K_P \tag{52}$$

assuming that $\frac{\partial \mathbf{K}_p}{\partial \theta_i} = \mathbf{0}$ for $i \neq p$. As this is a Kronecker product, the first term in (51) can be computed efficiently. The trace term in (51) can be computed by exploiting the cyclic property and the eigen decomposition as

$$\text{tr} \left( (\mathbf{K} + \sigma_n^2 \mathbf{I})^{-1} \frac{\partial \mathbf{K}}{\partial \theta_p} \right) = \text{diag} \left( (\boldsymbol{V} + \sigma_n^2 \mathbf{I})^{-1} \right)^T \text{diag} \left( \boldsymbol{Q}^T \frac{\partial \mathbf{K}}{\partial \theta_p} \boldsymbol{Q} \right), \tag{53}$$

where the latter term can be computed efficiently as

$$\boldsymbol{Q}^T \frac{\partial \mathbf{K}}{\partial \theta_p} \boldsymbol{Q} = Q_1^T K_1 Q_1 \otimes \ldots \otimes Q_p^T \frac{\partial K_p}{\partial \theta_p} Q_p \otimes \ldots \otimes Q_P^T K_P Q_P \tag{54}$$

and its diagonal as a Kronecker product of the diagonals of each factor in the product. For the noise parameter $\sigma_n$ we get $\frac{\partial (\mathbf{K} + \sigma_n^2 \mathbf{I})}{\partial \log \sigma_n} = 2\sigma_n^2 \mathbf{I}$ which makes both terms in (51) easy to compute.

Kronecker methods are also easily extensible for non-complete grids [7, 8] and non-Gaussian likelihoods [1].