[Reviews · NeurIPS 2017]

Reviewer 1



This paper proposes an interesting and new approach to learning flexible non-stationary (and I believe also non-monotonic, though this could be emphasized more) kernels. I have some concerns about the development and flow of the paper but overall I think it makes a useful contribution. Section 2.1 presents a bivariate version of the Spectral Mixture kernel, by inducing a dependence between s and s'. It would be helpful if a comparison could be made here to the work referenced later, by Kom Samo and Roberts. The remark at the end of 2.1 that the BSM vanishes presumably justifies your abandonment of this parameterization, but it would be helpful to have more exposition here; why doesn't this issue apply to the SM kernel (answer: because the SM kernel is stationary (?)). Are there no settings in which the kernel shown in Figure 1b would be a useful one? The kernel learned in Garnett, Ho, Schneider [ICML 2015] might be a setting, for example. Section 2.2 is the crux of the development but it very breezily goes from what seemed well motivated in Section 2.1 to a parameterization involving the Gibbs kernel and GPs giving smooth input dependent parameters for the BSM kernel. I would definitely appreciate more time spent here explaining these choices, especially in light of the remark that only comes at the end, which is that the solution to the spectral surface is unknown. Whether or not you thought it up this way, to me it seems that the motivation for the parameterization is that of making the parameters of the SM kernel input dependent rather than using the generalized Fourier transform. And ultimately, isn't the Gibbs kernel doing much of the work of non-stationary anyway? That needs more motivation. In 2.1 and 2.2 you remark that the extension to multivariate inputs is straightforward. If so, this should go in the Appendix. Eq (10) has an error: it should be log(w_ip), log(\mu_ip), and logit(l_ip), not w_ip, \mu_ip, and l_ip. Figure 5 is interesting. A numerical comparison would be useful as well---what is the MSE/log-likelihood for prediction (in-sample) and also forecasting (out of sample) using: BSM, GSM, SM, and SS? (A comparison to Kom-Samo would also be helpful, but I would very much like to see the others and I imagine they are more straightforward since they're all already implemented in GPML.) The SM kernel suffers from issues with initialization and local optima, especially with a large number of mixture components. It would be very useful to know whether your approach suffers from this issue as well.

Reviewer 2



--- Updated Comments Since Rebuttal --- This seems like very nice work and I hope you do keep exploring in your experiments. I think your approach could be used to solve some quite exciting problems. Some of the experiments are conflating aspects of the model (non-stationarity in particular) with what are likely engineering and implementation details, training differences (e.g. MAP), and initialization. The treadplate pattern, for instance, is clearly mostly stationary. Some of these experiments should be modified or replaced with experiments more enlightening about your model, and less misleading about the alternatives. It would also be valuable, in other experiments, to look at the the learned GP functions, to see what meaningful non-stationarity is being discovered. Your paper is also very closely related to "Fast Kernel Learning for Multidimensional Pattern Extrapolation" (Wilson et. al, NIPS 2014), which has all the same real experiments, same approach to scalability, developed GPs specifically for this type of extrapolation, and generally laid the foundation for a lot of this work. This paper should be properly discussed in the text, in addition to [28] and [11], beyond swapping references. Note also that the computational complexity results and extension to incomplete grids is introduced in this same Wilson et. al (2014) paper and not Flaxman et. al (2015), which simply makes use of these results. Wilson & Nickisch (2015) generalize these methods from incomplete grids to arbitrary inputs. Note also that a related proposal for non-stationarity is in Wilson (2014), PhD thesis, sections 4.4.2 - 4.5, involving input dependent mixtures of spectral mixture kernels. Nonetheless, this is overall very nice work and I look forward to seeing continued experimentation. I think it would make a good addition to NIPS. I am happy to trust that the questions raised will be properly addressed in the final version. --- The authors propose non-stationary spectral mixture kernels, by (i) modelling the spectral density in the generalized Bochner's theorem with a Gaussian mixture, (ii) replacing the resulting RBF component with a Gibbs kernel, and (iii) modelling the weights, length-scales and frequencies with Gaussian processes. Overall, the paper is nice. It is quite clearly written, well motivated, and technically solid. While interesting, the weakest part of the paper is the experiments and comparisons for texture extrapolation. A stronger experimental validation, clearly showing the benefit of non-stationarity, would greatly improve the paper; hopefully such changes could appear in a camera ready version. The paper also should make explicit connections to references [28] and [11] clear earlier in the paper, and should cite and discuss "Fast Kernel Learning for Multidimensional Pattern Extrapolation", which has all of the same real experiments, and the same approach to scalability. Detailed comments for improvement: - I didn't find there was much of a visual improvement between the SM kernel and the GSM kernel results on the wood texture synthesis, and similar performance from the SM kernel as the GSM kernel could be achieved for extrapolating the treadplate pattern, with a better initialization. It looks like the SM result on treadplate got trapped in a bad local optima, which could happen as easily with the GSM kernel. E.g., in this particular treadplate example, the results seem more likely to be a fluke of initialization than due to profound methodological differences. The comparison on treadplate should be re-evaluated. The paper should also look at some harder non-stationary inpainting problems -- it looks like there is promise to solve some really challenging problems here. Also the advantage of a non-stationary approach for land surface temperature forecasting is not made too clear here -- how does the spectral mixture kernel compare? - While the methodology can be developed for non-grid data, it appears the implementation developed for this paper has a grid restriction, based on the descriptions, experiments, and the imputation used in the land-surface temperature data. Incidentally, from an implementation perspective it is not hard to alleviate the need for imputation with virtual grid points. The grid restriction of the approach implemented here should be made clear in the text of the paper. - The logit transform for the frequency functions is very nice, but it should be clarified that the Nyquist approach here only applies to regularly sampled data. It would be easy to generalize this approach to non-grid data. It would strengthen the method overall to generalize the implementation to non-grid data and to consider a wider variety of problems. Then the rather general benefits of the approach could be realized much more easily, and the impact would be greater. - I suspect it is quite hard to learn Gaussian processes for the length-scale functions, and also somewhat for the frequency functions. Do you have any comments on this? It would be enlightening to visualize the Gaussian process posteriors on the mixture components so we can see what is learned for each problem and how the solutions differ from stationarity. One of the really promising aspects of your formulation is that the non-stationarity can be made very interpretable through these functions. - Why not perform inference over the Gaussian processes? - There should be more discussion of reference [11] on "Generalized spectral kernels", earlier in the paper; Kom Samo and Roberts also developed a non-stationary extension to the spectral mixture kernel leveraging the generalized Bochner's theorem. The difference here appears to be in using Gaussian processes for mixture components, the Gibbs kernel for the RBF part, and Kronecker for scalability. "Fast kernel learning for multidimensional pattern extrapolation" [*] should also be referenced and discussed, since it proposes a multidimensional spectral mixture product kernel in combination with Kronecker inference, and develops the approach for texture extrapolation and land surface temperature forecasting, using the same datasets in this paper. Generally, the connection to [28] should also be made clear much earlier in the paper, and section 4 should appear earlier. Moreover, the results in [*] "Fast Kernel Learning for Multidimensional Pattern Extrapolation", which have exactly the same real experiments, indicate that some of the results presented in your paper are from a bad initialization rather than a key methodological difference. This needs to be remedied and there need to be more new experiments. - Minor: "the non-masked regions in Figure 3a and 3f" in 5.2 is meant to refer to Figure 4.

Reviewer 3



This paper extends the spectral mixture kernel [28] to the nonstationary case, where the constant weights, length scales, and frequencies are replaced by nonconstant functions. The paper is well written and enjoyable to read. Here are some technical comments. 1. I do not quite understand the computational cost of evaluating the posterior and the gradient (stated to be O(PN^{(P+1)/P})), even after reading the appendix. The computation at least needs to perform P eigendecompositions on kernel matrices (size N*N for each data dimension). Therefore there should be at least a factor O(PN^3) in the cost? 2. Figures 2 and 3 show that the learned kernel is quite smooth, which must be because the learned functions w, ell, and mu are smooth. This is good but sounds a little mysterious to me. How does this happen? How would optimization end up with a smooth solution? 3. I suspect that the periodic pattern in the texture image 4(a) should be better modeled by using a periodic mean function rather than manipulating the covariance function. In such a case whether the kernel is stationary or not matters less.